# Healthcare professionals' views on the most important outcomes for non-infectious uveitis of the posterior segment: A qualitative study

**Mohammad O. Tallouzi** [1,2,3]*, **David J. Moore**[1], **Nicholas Bucknall**[4], **Philip I. Murray**[2,5], **Melanie J. Calvert**[1,3,6,7], **Alastair K. Denniston**[3,5,7,8], **Jonathan Mathers**[1,3]

1 Institute of Applied Health Research, College of Medical and Dental Sciences. University of Birmingham, Edgbaston, Birmingham, United Kingdom, 2 Birmingham and Midland Eye Centre, Sandwell and West Birmingham Hospitals NHS Trust, Birmingham, United Kingdom, 3 Centre for Patient Reported Outcome Research, Institute of Applied Health Research, College of Medical and Dental Sciences. University of Birmingham, Edgbaston, Birmingham, United Kingdom, 4 Patient Involvement Group in Uveitis (PInGU), Birmingham, United Kingdom, 5 Academic Unit of Ophthalmology, Institute of Inflammation and Ageing, College of Medical and Dental Sciences, University of Birmingham, Birmingham, United Kingdom, 6 NIHR Birmingham Biomedical Research Centre, NIHR Surgical Reconstruction and Microbiology Research Centre and NIHR Applied Research Collaboration (ARC) West Midlands at the University Hospitals Birmingham NHS Foundation Trust and the University of Birmingham, Birmingham, United Kingdom, 7 Birmingham Health Partners Centre for Regulatory Science and Innovation, University of Birmingham, Birmingham, United Kingdom, 8 Department of Ophthalmology, Queen Elizabeth Hospital Birmingham, University Hospitals Birmingham NHS Foundation Trust, Birmingham, United Kingdom

* m.tallouzi@bham.ac.uk, mohd.tallouzi@nhs.net

## Abstract

### Background

Uveitis comprises a range of conditions that result in intraocular inflammation. Most sight-threatening uveitis falls into the broad category known as Non-infectious Posterior Segment-Involving Uveitis (PSIU). To evaluate treatments, trialists and clinicians must select outcome measures. The aim of this study was to understand healthcare professionals' perspectives on what outcomes are important to adult patients with PSIU and their carers.

### Methods

Twelve semi-structured telephone interviews were undertaken to understand the perspectives of healthcare professionals. Interviews were audio recorded, transcribed and thematically analysed. Findings were compared with the views of patients and carers and outcomes abstracted from a previously published systematic review.

### Results

Eleven core domains were identified as important to healthcare professionals: (1) visual function, (2) symptoms, (3) functional ability, (4) impact on relationships, (5) financial impact, (6) psychological morbidity and emotional well-being (7) psychosocial adjustment to uveitis, (8) doctor / patient / interprofessional relationships and access to health care, (9) treatment burden, (10) treatment side effects, (11) disease control. Healthcare professionals recognised a similar range of domains to patients and carers but placed more emphasis on certain

**Data Availability Statement:** All relevant data are within the paper and its Supporting Information files.

**Funding:** This article represents an independent research project funded by the National Institute for Health Research (NIHR) under the Program Clinical Doctoral Research Fellowship Scheme at the University of Birmingham Award reference: CDRF-2014-05-057 (www.nihr.ac.uk). The views and opinions expressed by authors in this publication are those of the authors and not necessarily those of the NHS, the NIHR or the Department of Health. The funders had no role in study design, data collection and analysis, decision to publish, or preparation of the manuscript.

**Competing interests:** The authors have declared that no competing interests exist.

**Abbreviations:** BMEC, Birmingham and Midland Eye Centre; BUS, Birdshot Uveitis Society; COS, Core Outcome Set; HCP, Healthcare Professionals; HRQoL, Health-Related Quality of Life; PSIU-NRAS, Posterior Segment-Involving uveitis,National Rheumatoid Arthritis Society; PInGU, Patients Involvement Group in Uveitis; PPI, Patient and Public Involvement; UMO, Uveitic Macular Oedema.

outcomes, particularly in the disease control domain. In contrast the range of outcomes identified via the systematic review was limited.

## Conclusion

Healthcare professionals recognise all of the published outcome domains as patients/carers in the previous publication but with subtly differing emphasis within some domains and with a priority for certain types of measures. Healthcare professionals discussed the disease control and side effects/complications to a greater degree than patients and carers in the focus groups

## 1. Introduction

Uveitis is one of the leading causes of preventable blindness in the United States [1, 2], and accounts for approximately 10–15% of the causes of total blindness in Western countries and up to 25% of total blindness in the developing world [1]. Non-infectious intermediate, posterior and panuveitis are the most sight-threatening forms of uveitis. These forms involve the posterior segment of the eye and are often grouped together in clinical trials, named as Non-Infectious Uveitis affecting the Posterior Segment (PSIU) [3, 4]. Non-infectious uveitis may be purely ocular or associated with a systemic disease (e.g. ankylosing spondylitis, Behcet's disease or sarcoidosis [5, 6]. They may share common clinical features and potential complications e.g. macular oedema requiring additional treatment, either systemic or local injection-based therapy [7].

Trialists and clinicians aim to report certain measurements known as outcomes to evaluate new or existing uveitis treatment. However, inconsistency and heterogeneity in outcome reporting can hinder evidence synthesis that might prevent the evaluation and licensing of therapies by regulatory authorities [2] and misinform treatment decisions by clinicians and patients [8].

Healthcare professionals' and patients' perspectives have been a central element of patient's care in PSIU [9]. A recent systematic review illustrated the outcomes that have been used in clinical research [10], and recent qualitative research has also described patients' and carers' views regarding the outcomes that are important to them [11]. To our knowledge there has not been any qualitative research exploring healthcare professionals' views on important outcomes in PSIU. Furthermore, evidence from other diseases suggests that patients may differ in what they think is important compared to healthcare professionals [12]. Clinicians tend to evaluate and quantify diseases by assessing structural changes or direct measures of physiology, but these may not be meaningful to patients and carers, nor directly relate to the things that matter most to them e.g. aspects of function in daily life [13, 14]. It is therefore necessary to understand which outcomes matter to healthcare professionals and compare their views compared with those of patients and carers. This can be done within the context of the development of a core outcome set (COS) [15]. COSs are standardised sets of outcomes that have been scientifically agreed upon that are measured and reported in all trials for a specific clinical area that would promote consistency in outcome assessment [16]. Appreciating the different emphasis of healthcare professionals and patients will allow different stakeholder groups to appreciate where perspectives might differ, thereby encouraging inclusion of diverse opinions in core outcome set and other outcome-focused activities

Qualitative research approaches are often used in the initial stages of COS development to identify outcomes and outcomes domains that are important to key stakeholders, including patients and healthcare professionals [12, 17, 18]. As part of the study, "Development of a Core Outcome Set for Clinical Trials in Non-Infectious Uveitis affecting the Posterior Segment of the eye" [19], qualitative interviews were conducted to (a) understand the perspectives of healthcare professionals on what sort of outcomes they thought to be important for adult patients with non-infectious PSIU; and (b) contribute to the long-list of outcomes to feed into the Delphi exercise (an online survey used to seek participants' opinions were sought through two sequential rounds, with feedback from round 1 being provided. Participants would be asked to prioritise each outcome for inclusion in the study based on their level of importance using a nine-point Likert scale from 1 (no importance) to 9 (critically important); (c) help the development of a COS for use in efficacy or effectiveness trials in adult patients with PSIU. The views of healthcare professionals were combined with those of patients and carers. and we now report and compare the perspectives of healthcare professionals to the recently reported views of patients and carers with PSIU [11].

## 2. Methods

### 2.1 Study design

This is a qualitative research study using semi-structured telephone interviews. It was a component of a broader development of a core outcome set (COS) in noninfectious PSIU (COSUMO) [17, 20, 21]

### 2.2. Sampling

**2.2.1 Eligibility criteria.**   Participants were deemed eligible for telephone interviews if they were healthcare professionals (e.g. ophthalmologists, nurse practitioners) involved in caring for patients with PSIU, or health policy-makers/commissioners who were involved in commissioning with an influence on uveitis care or policy-making in the area of ophthalmology. All practitioners were urban-based consultant ophthalmologists with a special interest in uveitis tend to work in large eye units in major cities and patients are seen there rather than at smaller, peripheral, rural eye units

**2.2.2 Exclusion criteria.**   General practitioners and optometrists were excluded from the study. General practitioners have limited ophthalmological expertise and also lack the appropriate equipment to diagnose or manage patients with uveitis. Although the first presentation of patients with PSIU may be to optometrists, almost all ongoing care (other than refraction) is with the specialist eye services within a hospital ophthalmic department (comprising ophthalmologists and nurse practitioners), and the availability of those treatments is dependent on the wider health system (commissioners).

**2.2.3 Sampling.**   Participants were purposively sampled according to characteristics, including level of experience in ophthalmology and uveitis; level of experience in clinical trials; level of involvement as professional members of disease-specific patient groups; and across various NHS Trust sites in the UK. Sampling continued to a point of data saturation where no new outcomes or relevant concepts were being identified with further data collection i.e. a point of code saturation had been reached [22].

**2.2.4 Recruitment.**   Ophthalmologists specialising in uveitis, health policy-makers and health commissioners were recruited via UK and international clinical, research and health service networks, including the National Clinical Study Group (UK) and the International Uveitis Study Group. Nurse practitioners working with uveitis patients were invited via an International Ophthalmic Nurses Group. Eligible participants were identified by consultant

ophthalmologists (PIM, AD), contacted via email, and sent an invitation letter and a participant information sheet. The clinical doctoral research fellow (MOT) contacted all potential participants who responded, provided further information about the research and answered any questions prior to arranging an interview date. A convenient time and date were agreed, and a reminder was sent two days prior. Verbal consent was obtained from each participant before commencing the interview.

The sampling for this qualitative study was informed by a judgement of data saturation i.e. where the collection of new data no longer contributed additional elements to the outcome framework. Our conceptualisation of saturation was informed by the work of Cicely Kerr and colleagues specific to patient-reported outcome research [23].

### 2.3 Ethics

Ethical approval for the study was granted by the National Research Ethics Service (NRES) West Midlands–South Birmingham Research Ethics Committee (Reference number 17-WM-0111).

### 2.4 Data collection

Semi-structured telephone interviews with healthcare professionals were conducted to explore their views on the outcomes that are important to patients with PSIU. This method formed part of the wider study developing a COS in non-infectious uveitis of the posterior segment (PSIU study).

Interviews are a qualitative research data collection method that are widely used to gain in-depth knowledge and understanding of people's perceptions, attitudes, opinions, and experiences about disease and treatment [24]. Interviews are often used in the initial stages of COS development to identify outcomes and outcomes domains that are important to patients for any particular health condition [17]. Telephone interviews are introduced as a qualitative research method allowing researchers to have access to a wide geographical area and collect rich and in-depth data with evidence that they can produce equivalent data to face-to-face interviews [25, 26]

Telephone interviews were conducted by the clinical doctoral research fellow (MOT) between March and September 2018 and lasted from 34–52 minutes. Telephone interviews were audio-recorded and professionally transcribed. A topic guide was used which was refined iteratively during the first few interviews. Four topics were discussed in the interviews to understand healthcare professionals' perspectives regarding the most important outcomes that should be assessed in clinical trials focusing on PSIU. These topic areas highlighted healthcare professionals' thoughts on i) the impact of uveitis on patients' lives; ii) what outcomes are clinicians looking to achieve from the treatments provided for PSIU; iii) patients' expectations from treatment; IV) whether there were differences between patients' and clinicians' expectations. Prior to the start of interviews, all participants were asked to complete a short background questionnaire that gathered data on sample characteristics and helped to promote diversity amongst the interview sample (S1 Table)

The collection of patients' and carers' perspectives were fully described in the reference [11]. (Tallouzi, M.O., Moore, D.J., Bucknall, N., Murray, P.I., Calvert, M.J., Denniston, A.K. and Mathers, J.M., 2020. Outcomes important to patients with non-infectious posterior segment-involving uveitis: a qualitative study. BMJ open ophthalmology, 5(1), p.e000481.

### 2.5 Data analysis

Qualitative data were analysed using thematic analysis informed by the framework analytical approach [25, 27]. The framework has been used in our previous qualitative research work

informed by data collected from the focus group discussions describing outcomes important to patients with Non-infectious Posterior Segment-Involving Uveitis [11]. Interviews were digitally audio-recorded and transcribed clean verbatim by an external specialist transcription service. The clinical doctoral research fellow (MOT) checked the quality of the transcription and corrected any mistakes against the recordings by listening to each audio recording and reading the transcript repeatedly to become familiar with the data.

Initially, transcripts were read repeatedly to aid familiarisation and allow for data immersion. This facilitated the generation of preliminary codes and themes supported by the use of NVivo software, which eventually progressed into a developed coding frame.

Two authors coded percentage of the transcripts independently to cross-check the coding strategy. During this process our definition of an outcome was broad, including any consequence of PSIU or its treatment that clearly had significance to patients. Once we had finalised our coding framework it was then applied to the whole dataset (indexing). Data were then summarised descriptively retaining the original meaning of participants' discussions. Two of the most concept-rich transcripts were independently double-coded by two authors (MOT and JMM) and additional interpretations were incorporated into the coding frame. These early analytic findings were then discussed amongst the research team and agreement of the final themes was gained.

A coding framework of outcome domains and items had already been developed during the analysis of previous focus group discussions with patients with PSIU and their carers [11]. The focus group data were coded inductively (from the raw data), without application of any *a priori* outcome domain framework and then applied to the interview data where relevant. New codes were devised where healthcare professionals discussed outcomes and related concepts that had not been identified previously by patients and carers. A sample of interviews was coded independently by MOT and JMM and the results of this were discussed in order to refine the coding framework. The final analysis was discussed amongst the broader research team.

A comparison of the outcome domains was undertaken using items identified from the interviews with healthcare professionals with those previously identified via focus groups with patients and carers (8) and from a systematic review of the effectiveness of the pharmacological agents for macular oedema associated with non-infectious uveitis of the posterior segment [10].

## 3. Results

### 3.1 Sample characteristics

A total of 15 participants were approached (7 ophthalmologists, 5 nurse practitioners and 6 health policy-makers/commissioners), and of those 12 (80%) agreed to take part. Telephone interviews included 5 ophthalmologists, 3 nurse practitioners and 4 health policy-makers/commissioners. The mean ophthalmology experience was 14 years. The majority (9 interviewees) reported their involvement in clinical trials of uveitis, and 6 interviewees expressed their involvement in disease specific patient groups such as the Patient Involvement Group in Uveitis (PInGU) West Midlands, the Birdshot Uveitis Society, Behcet's UK and the National Rheumatoid Arthritis Society (NRAS). Further details of participants' characteristics are reported in Table 1

### 3.2 Identification of outcome categories and core outcome domains

A total of 43 separate outcomes were interpreted and mapped onto the 11 previously reported outcome domains [11]: (1) visual function, (2) symptoms, (3) functional ability, (4) impact on relationships, (5) financial impact, (6) psychological morbidity and emotional well-being, (7) psychosocial adjustment to uveitis, (8) doctor/patient/inter-professional relationships and

**Table 1. Interviewee details.**

| No | Job Role | Years of experience as Consultant Ophthalmologist | Contributed to clinical trials (y/n) | Experience in ophthalmology commissioning (y/n) | Involvement in patient groups (y/n) |
|---|---|---|---|---|---|
| 1. | Ophthalmologist | 20 | Yes | Yes | Yes |
| 2. | Ophthalmologist | 12 | Yes | No | No |
| 3. | Ophthalmologist | 4 | Yes | No | Yes |
| 4. | Ophthalmologist | 5 | Yes | No | No |
| 5. | Ophthalmologist | 33 | Yes | Yes | Yes |
| 6. | Nurse practitioner | 24 | Yes | No | Yes |
| 7. | Nurse practitioner | 5 | No | No | Yes |
| 8. | Nurse practitioner | 5 | No | No | Yes |
| 9. | Health policy-makers/ commissioners | 25 | Yes | No | Yes |
| 10. | Health policy-makers/ commissioners | 10 | No | Yes | No |
| 11. | Health policy-makers/ commissioners | 20 | Yes | No | No |
| 12. | Health policy-makers/ commissioners | 5 | Yes | Yes | Yes |

access to health care (Service outcomes), (9) treatment burden, (10) treatment side effects, and (11) disease control. Table 2 described in details outcome domains as discussed by healthcare professionals, however Table 3 provides a comparison of the outcome domains and items identified from the interviews with healthcare professionals with those previously identified via focus groups with patients and carers [11] and a systematic review of the effectiveness of the pharmacological agents for macular oedema associated with non-infectious uveitis of the posterior segment.

**3.2.1 Outcome domain 1: Visual function.** Healthcare professionals (HCPs) used vision as a broad term comprising distance vision, colour vision, peripheral vision and contrast sensitivity, and used the terms vision and visual function interchangeably. Although healthcare professionals identified distance vision as a basic test in ophthalmology practice and widely used in clinical trials, they emphasised that other important elements of visual function should be considered prior to making decisions for uveitis treatment. Whilst there are limited measures of vision in clinical practice there is recognition amongst at least some interviewees that visual function is broad and multi-faceted:

> "So when I say vision, usually this means Snellen vision in clinical practice, but vision means more than just Snellen vision, it means visual acuity, it means field of vision, it means contrast sensitivity, it means colour vision, it means all these other things, lack of distortion and so on, reading speed, all these things. So vision is a broad term that can mean different things for different patients. So acuity is the bread and butter visual outcome, but there are other things that are important when measuring visual outcome." HCP 2

Healthcare professionals noted that they may prioritise treatment according to easily measurable outcomes such as distance vision and presence of macular oedema, and the need to recognise that these might not be as meaningful to patients as other outcomes:

> "I think [as] clinicians we go very much on the visual acuity and the presence or absence of macular oedema or signs of inflammation in the eye. So there is a temptation to treat what we

**Table 2. Outcome domains and items identified from the interviews with healthcare professionals.**

| No | Outcome domains | Definition of domain | Items in the domain |
|---|---|---|---|
| 1. | **Visual function** | Describes the impact of PSIU on aspects of patients' vision | Distance vision, near vision, contrast sensitivity, colour vision, peripheral vision |
| 2. | **Symptoms** | Describes patients' bodily experiences that result from PSIU | Painful eye, photosensitivity, redness, floaters, visual disturbance, distortion of vision |
| 3. | **Functional ability** | Describes the impact of NIU-PS on patients' ability to perform, maintain or continue their day-to-day functions | Work/employment (maintaining / adjustments), driving/commuting related impact, activities of daily living and self-care, participation in social and leisure activities |
| 4. | **Impact on relationships** | The impact of PSIU on relationships with others | Intra-family and spousal relationships; friendships |
| 5. | **Financial impacts** | Describes the financial impacts of having PSIU | Financial cost to patients due to work loss, early retirement and other treatment related cost (e.g. travelling cost) |
| 6. | **Psychological morbidity and emotional well-being** | Describes the psychological and emotional morbidity that may occur in patients with PSIU | Depression and mental illness, anxiety, stress, emotional well-being |
| 7. | **Psychosocial adjustment to uveitis** | Describes how well people with uveitis adjust to life with the disease and how it influences self-image. This partly results from day-to-day interactions with others e.g. family, friends, and other people. | Threats to psychosocial well-being, coping strategies, Indicators of psychosocial adjustment (sense of normality) |
| 8. | **Doctor/patient/inter-professional relationships and access to health care (Service outcomes)** | Describes the communication between doctor and patients; the ability to access uveitis clinics and uveitis care facilities | Clinician-patient relationship, shared decision-making, access to physical aids and other resources, access to counselling and psychotherapy services |
| 9. | **Treatment burden** | Describes the work that people with uveitis need to do to care for their health and its effect on their life. | Number of hospital visits, amount of medication, adherence and tolerability |
| 10. | **Treatment side effects** | Describes undesired effects of the treatment | Treatment side effects (ocular and systemic) |
| 11. | **Disease control** | Describes how to control PSIU | Anterior segment activity (cells, flares); vitreous activity (cells, haze); retinal vasculitis; retinitis; raised intraocular pressure; macular oedema; cataract; other ocular comorbidities; prevent disease progression and long-term damage including retinal scar/atrophy/ischaemic, optic atrophy and prevent Flare/relapse/recurrence |

see on imaging and what we see on a visual acuity measurement, whereas for the patient they might not really see that in the same way. They might be more concerned about how they feel, the side effects of the steroids than what their visual acuity is." HCP 3

Visual function was identified as an important outcome domain by the majority of interviewees and was considered by them to be associated with many other outcome domains including patients' functional ability e.g. working, driving, and daily activities, relationships, financial impact, psychological and emotional wellbeing, and psychosocial adjustment to uveitis:

*"There is a huge impact on the dynamics of day-to-day, so those people will first be worried about their visual loss, it may change their mood, it may change the way they interact with the rest of the family, with people around them in general, they will have to be brought sometimes to hospital by someone else so someone has to take time off work and come with them. There will be an impact if they can't work properly, so they are not going to make money, so there is a financial consequence to the family. So there are many angles of the care of these patients and the disease itself that can impact on the people around the patient."* HCP 5

**3.2.2 Outcome domain 2: Symptoms.** A wide range of sensory-related bodily experience was discussed in the interviews (e.g. pain, red eye, visual disturbance) that may have a significant impact on patients' functionality:

**Table 3. Comparison of outcome domains and items between professional interviews, focus groups with patients and carers [11] and outcomes assessed in clinical research identified via systematic review [10].**

| Outcome domains and items | Healthcare Professionals | Patients/carers [11] | Systematic review [10] |
|---|:---:|:---:|:---:|
| **Visual function** | | | |
| Distance vision | ✓ | ✓ | ✓ |
| Near vision | ✓ | ✓ | |
| Contrast sensitivity | ✓ | ✓ | |
| Colour vision | ✓ | ✓ | |
| Peripheral vision | ✓ | ✓ | |
| Depth perception | | ✓ | |
| **Symptoms** | | | |
| An uncomfortable or painful eye/s | ✓ | ✓ | |
| Photosensitivity | ✓ | ✓ | |
| Redness | ✓ | ✓ | |
| Floater | ✓ | ✓ | |
| Visual disturbance | ✓ | ✓ | |
| Distortion of vision | ✓ | ✓ | |
| Fatigue | | ✓ | ** |
| Watery eye | | ✓ | |
| Headache | | ✓ | |
| **Functional ability** | | | |
| Work/employment (maintaining / adjustments) | ✓ | ✓ | |
| Driving/commuting related impact | ✓ | ✓ | |
| Education related impact | | ✓ | |
| Activities of daily living and self-care | ✓ | ✓ | |
| Participation in social and leisure activities | ✓ | ✓ | |
| **Impact on relationships** | | | |
| Intra-family and spousal relationships; friendships | ✓ | ✓ | ** |
| **Financial impacts** | | | |
| Financial cost to patients due to early retirement, the need to take a part-time job or redundancy | ✓ | ✓ | |
| Financial cost to patients due treatment-related cost (e.g. travelling cost) | ✓ | ✓ | |
| **Psychological morbidity and emotional well-being** | | | |
| Depression and mental illness | ✓ | ✓ | ** |
| Anxiety | ✓ | ✓ | |
| Stress | ✓ | ✓ | |
| Frustration and Anger | | ✓ | |
| Emotional wellbeing | ✓ | ✓ | |
| **Psychosocial adjustment to uveitis** | | | |
| Threats to psychosocial well-being | ✓ | ✓ | |
| Coping strategies | ✓ | ✓ | |
| Indicators of psychosocial adjustment (sense of normality, sense of self and identity) | ✓ | ✓ | |
| **Doctor/patient/inter-professional relationships and access to health care** | | | |
| Clinician-patient relationship/communication | ✓ | ✓ | |
| Inter-professional relationships | | ✓ | |
| Shared decision-making | ✓ | ✓ | |
| Access to uveitis clinics and/or facilities | | ✓ | |
| Access to physical aids and other resources | ✓ | ✓ | |
| Access to counselling and psychotherapy services | ✓ | ✓ | |

*(Continued)*

**Table 3.** (Continued)

| Outcome domains and items | Healthcare Professionals | Patients/carers [11] | Systematic review [10] |
|---|:---:|:---:|:---:|
| **Treatment burden** | | | |
| Number of hospital visits | ✓ | ✓ | |
| Amount of medications | ✓ | ✓ | |
| Adherence | ✓ | ✓ | |
| **Treatment side effects** | | | |
| Treatment side effects (ocular and systemic) | ✓ | ✓ | ✓ |
| **Disease control** | | | |
| Anterior segment activity (cells, flares) | ✓ | * | ✓ |
| Vitreous activity (cells, haze) | ✓ | * | ✓ |
| Retinal vasculitis | ✓ | * | ✓ |
| Retinitis | ✓ | * | ✓ |
| Raised intraocular pressure | ✓ | ✓ | ✓ |
| Macular oedema | ✓ | ✓ | ✓ |
| Cataract | ✓ | ✓ | ✓ |
| Other ocular comorbidities | ✓ | ✓ | |
| Prevent disease progression and long-term damage including retinal scar/atrophy/ischaemic, optic atrophy | ✓ | ✓ | |
| Prevent Flare/relapse/recurrence | ✓ | ✓ | ✓ |

* Patients discussed inflammation but didn't use these specific terms to reflect inflammatory markers and retinal inflammation highlighted by the healthcare professionals

** A single study stated that these components of HRQoL were assessed using the SF-36, but no further information was provided.

*"I think obviously patient symptoms are important: if they have pain, inflammation, red eye, floaters, blurred vision, poor vision. The aim is to be symptom free."* HCP 7

Clinicians thought the impact of floaters varied between patients depending on the type of work patients have to carry out. For patients whose quality of vision was affected by floaters impacting on their day-to-day activities and professional work, interventions were needed to resolve this issue and enable them to function properly. Thus floaters could impact patients' functional ability e.g. ability to maintain and continue working:

*"So take for instance the issue of floaters. Now most of the time we as clinicians ignore floaters, but for some patients they're really important. . . so what I'm thinking here is maybe a professional squash player, suddenly sees ten balls coming at him at once rather than one. Or the worst one I had was actually an ornithologist. His job was to stand in fields and count the number of birds in a flock; and you can imagine with a whole load of floaters what a problem that was. And because he brought this up, this was the one time or one of the few times that I've actually recommended that these patients actually have a vitrectomy. So there are instances where the patients will bring up something that's seriously important to them or really important to their working lives and of course we will pay attention to that."* HCP 1

**3.2.3 Outcome domain 3: Functional ability.**    Healthcare professionals discussed the impact of uveitis on patients' functional ability in a wide range of aspects e.g. working, driving,

carrying out normal daily activities and participating in social and leisure activities). Healthcare professionals linked patients' functional ability to quality of life:

*"Uveitis can have a profound effect on quality of life. If it's somebody of working age, if they can no longer competently do their job, if they can't. . . if they have to drive to work and they can't drive any longer. I think it can affect all aspects of their lives really. Yeah, even as a daily chore to going to work or looking after the kids, or if they have got carers it does impact a lot. If they can't work they will not bring salary home and that will make their life even worse than when they're working."* HCP 3

Healthcare professionals described uveitis as a debilitating disease that leaves a huge impact on people's ability to engage in usual daily activities including cooking, cleaning and shopping. Being unable to carry on such day-to-day activities puts a significant burden onto their carers. They also highlighted that it may be unsafe for patients to perform such activities:

*"So for example the need for a carer to take a day off and come to the clinic with the patient, the need to do all the cooking and the cleaning and the shopping at home because it may not be safe for their partner to do it, so an increased level of domestic responsibility."* HCP 5

Healthcare professionals linked uveitic macular oedema to patients' visual function leading to a negative impact on patients' functional ability (working, driving). In consequence patients suffer financial difficulties due to loss of work or redundancy:

"*Then those patients with macular oedema who perceive things as looking through a mist and are unable to read, that's a huge limitation for them in terms of their work, so they are not able to work effectively. We have a disproportionate number of people who have to miss work because of their inability to see properly for periods of time while they are being treated for ocular inflammation."* HCP 7

Healthcare professionals made a point that functional ability is influenced by visual function and may need further intervention including psychological, social and physical action to be in place:

*"It's the visual function that affects the function of your life when we're talking about uveitis, particularly if the vision has fallen to such an extent that the person can't do their job, can't drive or can't enjoy life, or needs additional caring. You can't see, so yes at the end of the day it all comes back to vision."* HCP 1

**3.2.4 Outcome domain 4: Impact on relationships.** The impact of PSIU and subsequent treatment on family relationships and future family unity was discussed in the interviews. This was linked to the degree of visual loss and the level of dependence that patients may end up with. Thus the impact of PSIU was a major reason to create tension between family members:

*"Yes, I would think extremely important, particularly if the vision has fallen to such an extent as the person can't do their job or can't enjoy life, or needs additional caring and what have you, that's going to have a major impact of the if you like the family unit or whoever is closest to the patient themselves. They may lose their job or be unable to work or then their partner*

*will have to support them or their families will have to support them which can lead to lots of tension.*" HCP 1

Furthermore, healthcare professionals discussed the impact of uveitis on patients' ability to look after children and the role of parenthood and the relationship between a mother/father and children:

"*I think it goes broader than just the direct family, I think there are ripples beyond, if you're a mother then you have children and those children can be impacted, so it's not necessarily just the person that turns up to the clinic, it can be on other people who don't get what's going on.*" HCP 11

Interviewees linked spousal relationships to patients' financial and functional ability. Therefore being unable to work and earn income impacts upon the family relationships and unity:

"*Even relationships: relationship is a big role where your eyes are concerned, and there are a lot of things you can't do that you could do before, so you would be depending on a carer or your partner. Basic things, relationship-wise things they could do before, their partner might feel they are a burden on them, so they have to work things around the patient themselves. So the partner might feel frustrated themselves that they are doing so much, or they might get stressed out themselves and doing too much and get burnt out.*" HCP 7

**3.2.5 Outcome domain 5: Financial impact.** The financial impact of PSIU was highlighted by healthcare professionals and perceived as an important multi-faceted outcome. For example, loss of work was one of the major issues that negatively impact the financial status of patients and their families. Consequently, interviewees felt that financial struggles could negatively impact patients and family relationships:

"*It [uveitis] can have an effect if the person is not able to work, it can affect the income coming into the family, so they could have financial concerns about their mortgage, even as a daily chore to going to work or after looking after the kids, or if they have got carers it does impact a lot. If they can't work they will not bring salary home and that will make their life even worse than when they're working.*" HCP 6

Interviewees noted that socio-economic background may affect the extent to which people are financially impacted by uveitis, for example through the need to take time off of work to attend uveitis speciality clinics or prescription costs, travel costs and frequent hospital visits:

"*Pretty much* [for] *any working class* [person] *having to visit the hospital and having to take medication, will have an economic impact. Obviously whatever the environment they have worked in, often they may not get time off to come where they need to come and they may have to lose their pay, or they may have to go off sick for a while which all will have an impact on their work life balance and their economy.*" HCP 4

Furthermore, the financial burden was also discussed in relation to psychological and emotional illness (e.g. depression, anxiety), thus the psychological well-being and family relationships (family unity) are influenced by the financial impact:

*"It's all interconnected really. It's all part of the same problem; I think it has lots of connotations in terms of how a partnership or a family unit, the dynamics of a relationship or a family unit would work. If there's things like loss of income, or there's one of the parents is actually very depressed, of course that's going to have a big impact upon the way that unit works together and lives together."* HCP 6

**3.2.6 Outcome domain 6: Psychological morbidity and emotional well-being.** The impact of PSIU on patients/carers psychological and emotional well-being was frequently highlighted among interviewees, especially ophthalmologists. Psychological impacts occur due to loss of vision or fear of loss of vision. Psychological morbidity and emotional well-being are broad and multi-faceted including depression, stress, frustration and anxiety:

*"Uveitis can cause a mental problem as well. Depression from being unable to see and vision being blurred and poor vision, it causes patients to be depressed. They are very anxious and depressed very often because they're scared that they will never be able to see again or they might go blind; their partner often is involved with that."* HCP 7

Participants discussed how psychological and mental health are now well-recognised in patients with PSIU and that some ophthalmologists now include psychology support services within specialist uveitis clinics. In these cases patients may have an initial psychological assessment in the uveitis clinic and then be referred to the psychology support service if ophthalmologists feel further care is needed:

*"I think I have seen that many patients, once they have a diagnosis of uveitis their lives change, they become focused on that, it takes over, and that's the reason why we developed these psychological or a psychology support service, because I felt that many patients were trapped into this, and they would continue their normal life, they were taken over."* HCP 5

**3.2.7 Outcome domain 7: Psychosocial adjustment to uveitis.** The psychosocial impact of uveitis was obviously significant and discussed in most of the interviews. Related to this, interviewees covered various aspects of psychosocial adaptation/adjustment to PSIU describing patients' ability to adjust their lives with the disease and how it affects their sense of normality. Three components were identified in this domain (1) threats to psychosocial well-being–the things that indicate that patients are having difficulties with adjustment; (2) coping strategies–the strategies that people use in order to master, tolerate, or reduce the impacts PSIU on psychosocial issues; and (3) indicators of psychosocial adjustment. Related to this, interviewees discussed various threats to psychosocial well-being, for example, a feeling of loss of autonomy and independence:

*"It all relates back to I think the patient's ability to lead an independent life so that they can lead the life that they want to lead as best as they can do, and live with the disease rather than have their life dictated to them by the disease. Thus to cope with it, so they can manage their lives in such a way that their eye disease is a secondary thing, it's not impacting on their ability to move forward, new careers or their present career that they can. . . I think for a lot of them it reverts back to a facsimile of a life that they used to have prior to their diagnosis."* HCP 5

Another threat to psychosocial well-being discussed in the interviews was lack of predict-ability of the disease and impacts and related uncertainty regarding the future of the disease:

*"We know that we cannot always give them a name, [or] that we can give a name that doesn't mean much as a descriptive thing and we don't really understand many times why they are having it. So we can say 'I can see the signs of the disease, I know where it's affecting your eyes but I don't know exactly why it's happening', and the second aspect is that we are realistic about knowing that what I want to do is prevent visual loss, but knowing that there is many times a limitation to that, and I can't just switch it off, I can't just cure them completely."* HCP 10

In response to those threats, interviewees pointed to the importance of the coping strategies that patients with PSIU use to enable them to master, tolerate, or reduce the impacts of PSIU on psychosocial issues. These may include a mix of psychological e.g. acceptance of NPSIU, positive attitudes, psychological support and behavioural strategies e.g. changing diet, stopping smoking or modifying day-to-day routines:

*"I suspect things that might help is modifying their lifestyle . . . a very common question from patients is 'Is there anything I can do? Is there any diet I can follow? Anything like that will help improve this disease?' I normally tell patients to go away and try [it], because who knows? Maybe they do. Additionally. . . it's well known that if people smoke it makes macular oedema worse, so we always advise against that sort of thing. And then there are a small, I have to say a very small group of patients, where there are lifestyle events which seem to trigger off relapses of uveitis."* HCP 1

Interviewees felt that psychosocial adjustment is crucial in patients with PSIU and is highly linked to patients' psychological and emotional support which helps to promote positive cop-ing mechanisms and helps to strengthen patients' ability to accommodate the disease:

*"I think it's about facilitating support for that patient and not just expecting them to go out and find it themselves. For instance, if they needed any support with employment, if they needed some advice or they their employers needed advice about what they could do to help, say the person has got problems using a computer and they need some support with that, some new equipment, and it's about putting them in touch with the right people and helped them to gain support, and that's what we would do in the hospital. So it's just making sure that when patients leave us they have contact numbers, they know that if they run into any problems that they have got someone they can call, they've got rapid access to hospital services, those sorts of things. I think there are some patients who really find it very difficult to cope with, and I think psychology support is essential."* HCP 6

Interviewees also talked about indicators that people with uveitis have gone through pro-cesses of psychosocial adjustment. Related to this, this interviewee discussed patients' percep-tions of regaining vision and a sense of normality:

*"The expectation is to return back to normal. I mean your ability to see clearly enough to enable you to conduct your activities of daily living, the standard which you would deem nor-mal for you. [This] could encompass field of vision (a proportion of patients have reduced field of vision); it could involve central vision acuity; it could involve near vision acuity, [or] dis-tance visual acuity. These are all parameters of visual function."* HCP 4

**3.2.8 Outcome domain 8: Doctor/patient/inter-professional relationships and access to health care.** This domain describes the quality of the relationship between healthcare professionals, especially ophthalmologists, and patients, as well as inter-professional relationships that facilitate effective care provision. One ophthalmologist described how important it is to build a trusting relationship with patients and keep them well informed during disease progression to create a better understanding of their situation. One interviewee described a process of showing patients their eye scans and taking them through any changes providing comparison before and after treatment. The aim of this approach was to help patients have a better understanding of their eye condition, create positive engagement, enhance the patient and clinician relationship and encourage clinicians to listen to patients' concerns and try to help them:

> "*I always like to share what I see with the patient, so if I'm getting the results back and I can see what's happening, I try to explain to them what we are achieving with those steps that we are doing. So they have a feeling for 'yes my disease is changing'. So if you do a visual field . . .and the patient's fields are getting better you share that with them. You explain to them. It's easier for them to be reassured that something positive is happening even though they are not totally aware of it; that things are not getting worse or that they are getting some improvement. So it's important that this information, these things are shared with the patient and explained to them.*" HCP 5

In describing the doctor/patient relationship patients' engagement in treatment decision-making was discussed. This participant expressed some degree of uncertainty around how much engagement in shared decision-making; some patients desired and expressed a need for both parties to understand the other's perspective:

> "*We don't communicate with patients particularly well, and I don't know how engaged or how much they want to be engaged with understanding what our outcomes are. . . shared decision-making is a great expression, but it's tough to make shared decision-making when neither understand the other's perspective. So that involves a certain amount of understanding and also an understanding of the willingness of the patient to understand the medical perspective, but also willingness of physicians to take heed of the patient perspective.*" HCP 3

Healthcare professionals expressed the importance of access to healthcare facilities including psychological, social and practical facilities e.g. visual aids. Ophthalmologists emphasised the need to inform patients about available support services (psychological, low vision aids etc.) and advise them on the referral process to the specialist support services where appropriate. For example, ophthalmologists advised on available psychotherapy and counselling services and the process of including patients in this service which has become a part of the uveitis care service in a few NHS Trusts. The emphasis given to this topic by the participants indicates the importance and the role of psychological support:

> "*We created a group of psychologists; we have two psychologists working with us who we would flag to them anyone in clinic who we felt needed to talk to them and we offer that to patients and the patients would have a chance to speak to them. They would assess the patients, they would decide if anything beyond that needs to be done so they could escalate the care towards something higher like a senior psychiatrist or any help, mental health groups in the location when they leave. [We] involved them with that intention because we knew that*

*these patients were struggling. Many of them were struggling very badly with the disease, with the management, with everything, and they had no one to talk to. In clinics we are not equipped to talk, we don't have the time, and most of the patients leave frustrated because they get the prescriptions, the instructions and all that but they haven't had a chance to talk to anyone about their problems. That's why I fought very hard to introduce this.*" HCP 5

Further discussion focused on accessing visual support services that aim to provide visual aids and support patients with PSIU to live their lives:

*"I meant referring them to visual support services, whether that's the RNIB, whether it's Action for Blind People, that kind of thing. So it's about facilitating that and just enabling that process. There's RNIB, they have got loads of aids, walking stick, and they gave gadgets that when you are pouring a drink it tells you when to stop. There's the scanner they use where you can scan a newspaper and it projects on the TV, things like this. There's loads of gadgets that you use now, people have been using in trial, those glasses that can detect some motion.*" HCP 6

**3.2.9 Outcome domain 9: Treatment burden.** Treatment burden is the workload of healthcare undertaken by patients and carers. Healthcare professionals described the amount of effort that patients are required to make in order to manage their health condition. Interviewees highlighted a significant treatment burden as a result of frequent hospital visits, amount of medication and adherence to the treatment. Thus treatment burden could hamper patients' general well-being and day-to-day life:

*"Well I suppose another thing might be they might start to feel a bit institutionalised because they're having to come to hospital a lot, and being checked. . . frequent hospital visits, I think that can be difficult.*" HCP 6

Additionally, healthcare professionals expressed a proportional relationship between frequency/intensity of symptoms and the amount of effort needed to manage those symptoms including frequent hospital visits and additional treatment:

*"I think obviously patient symptoms are important, if they have pain, if they have visual quality of life issues, it will impact their life in-terms of how many times they have to go to the hospital, how many times they have to inject the medicines, how many times they have to see the doctor.*" HCP 10

Additionally, healthcare professionals discussed the impacts of treatment on patients' psychosocial status and the routine they have to establish around the treatment in order to lead a normal life:

*"So taking the drugs it's not easy. People who don't need to take medication don't realise the burden of having to take medications. There is a burden to the body and to the mind of having to do that, every day remembering you have to take tablets in the morning, in the middle of the day, at night, two drops, so their lives change, they cannot run a normal life anymore. If you are going to travel they have to remember to bring all the medication with them, they are afraid of relapses when they are going away, all this is stuff that we don't think about but they think about.*" HCP 5

**3.2.10 Outcome domain 10: Treatment side effects.** Discussion with healthcare professionals also focused on the safety aspects of medication and their potential side effects. Interviewees referred to reducing or having no side effects as a desired outcome. Consequently, clinicians are working with their patients to reduce symptom and side effects:

*"Everything needs to be done within the safety of the patient's general health, so you have patients who because of other problems to their health will not tolerate some treatments well or have more side effects or will really have problems that will stop some drugs being used, and so the outcome is apart from all this making sure safety of the patient, safety on the way you manage the patient, you don't induce any more complication."* HCP 5

Healthcare professionals reported two main categories of side effect comprising ocular and systemic aspects as described in the following quote:

*"Aspects like cataracts or intraocular pressure rises that could be an adverse effect, corneal health, retinal detachment, any of these that could be safety issues-related to the treatment should be captured. From my experience the most common morbidity and side effects are due to long term steroids, so the side effects and adverse effects of long-term steroids are significant. So I think osteoporosis, diabetes, weight gain, stomach problems, infections, hypertension, skin, all these side effects of steroids are probably cumulative and much greater than the side effects from people on immunosuppression."* HCP 2

Further discussion highlighted potential effects on the psychological and emotional status of patients, especially of corticosteroids:

*"The treatment itself, especially steroids can make them have psychological side effects which we completely under acknowledge, so they will be stressed, they will be sleepless, they will be either very happy or very sad, and then their partner is the one who has to tolerate it."* HCP 7

Side effects were associated with treatment burden by which additional hospital visits, admissions and further treatment are required. Interviewees felt this is very costly to the patient in ways other than money e.g. health issues:

*"The cost I mean the side effects, all the damage that they can suffer from the treatments we are giving. So patients who develop renal problems because of the drugs we use, or become diabetic, or the diabetes develops further, anything that is related to the treatment, exposure to the treatment which can cause health issues is a price you are paying, so they are paying a price for that, it's a question of how high is that price will be. So if you think they are okay with a low level of medication, controlling the eye problem and not suffering anything wonderful."* HCP 5

Furthermore, side effects negatively impact patients' functional ability and psychological and emotional well-being. This encompasses time lost from work and other day-to-day activities; as a result quality of life can be affected:

*"There is an aspect of quality of life that I didn't mention and that's side effects. Some patients simply cannot get along with the medication that we offer them, and are debilitated by side effects. [It is] either their visual function or to be distraught by side effects. [This] also*

*constitutes a problem for their quality of life. Quality of life means the ability to conduct their daily activities that exist without a feeling of lack of wellness. So that lack of wellness could be manifested by systemic symptoms, so for example: headache as a result of side effects, anxiety due to steroids, or depression induced by steroids. These are all physical symptoms which impact in terms of wellbeing. So I'm talking about a feeling of wellbeing."* HCP 3

**3.2.11 Outcome domain 11: Disease control.** Uveitis is essentially an inflammatory process within the eye, thus clinicians' main concern was to stop or control inflammation. This was explored in detail by healthcare professionals, especially ophthalmologists:

"*Chronic recurrent inflammation is damaging to the eye, so your objective is to control inflammation the best you can and prevent the recurrences of the disease in the future. So it prevents progression to a chronic disease or prevents recurrences that could be damaging. It's very intense. So I think objective is to try to improve inflammation, prevention of recurrences and hopefully prevention of visual loss.*" HCP 5

Ophthalmologists highlighted how important it is to stop the disease process and prevent disease progression to preserve patients' vision where possible:

"*The aim is to stop the disease process itself; often it's not possible in uveitis, and particularly in the conditions like idiopathic uveitis. Often you have to deal with patients balancing the treatment and lifestyle issues. [*The aim is to*] balance between good control of the disease, less side effects in the medication patients have to take, less visits a patient has to make to the hospital, and create a good work life balance.*" HCP 4

Disease complications were also perceived as an important outcome in the process of controlling the disease and its progression. Ophthalmologists described an association between visual function and retinal complications. For example, uveitic macular oedema (UMO) is associated with poor distance vision and reading vision which can consequently impact patients' functional ability (e.g. driving, working, reading, computer work and day-to-day activities). Furthermore, retinal vasculitis/retinitis describes an inflammation either to the retinal blood vessels or to the retina and is also linked to patients' peripheral vision. Therefore assessing patients' visual field is essential in those patients:

"*Visual function was correlated to the retinal findings. For example drop in visual acuity is linked to macular oedema. So for example if you've got [some forms of] posterior segment involving uveitis there may be no retinal vasculitis but there is macular oedema, in which case you should select pure visual acuity and macular thickness [as your outcome measures]. If you have retinal vasculitis you would select visual field for example as one of your [outcome] measures, and that's what your index consists of. You have a minimum of two or three features within the index that you target for use, or for your patients, because clearly not everyone will have all aspects of disease who have expected to have inflammation.*" HCP 3

Furthermore, ophthalmologists described various measures to assess the disease activity including inflammatory markers e.g. anterior chamber activity, vitreous activity or vitreous haze and structural changes e.g. macular oedema, epiretinal membrane. However, it is most important to link those findings to patients' functional ability and visual function:

*"Certainly any marker of inflammation that can be linked to patient benefit would certainly be considered: change of anterior chamber inflammation grade, change in vitreous inflammation or vitreous haze, change in visual acuity. So I think anatomical measures are potentially relevant if they are supportive of its physiological parameters and if they can be linked to patient benefits and quality of life."* HCP12

Further discussion was constructed in regard to flare up and relapses. Relapse was also associated with visual function and therefore was considered as an important outcome in controlling uveitis:

*"So one definition might be a relapse that requires an increase in systemic steroid dose for example. So for that patient each time they have a relapse they need the steroid dose to go up, so the cumulative steroid dose over one year if they have two or three relapses is high. So the outcome in that patient would be the aim would be to reduce the number of relapses to zero so you don't need to have recurrent flare ups. . . recurrent doses of steroids. Then there's each relapse is associated with a reduction in vision, then the aim would be to prevent any relapses that's reducing their vision, so they have good vision throughout the year instead of every three months they can't see properly."* HCP 2

### 3.3 Comparison of healthcare professionals views on outcomes with (a) patients and carers views and (b) findings of a systematic review of the effectiveness of pharmacological agents for macular oedema associated with non-infectious uveitis of the posterior segment

Table 3 provides a comparison of the outcome domains and items identified from the interviews with healthcare professionals with those previously identified via focus groups with patients and carers [11] and a systematic review of the effectiveness of the pharmacological agents for macular oedema associated with non-infectious uveitis of the posterior segment [10].

Table 3 highlights the fact that healthcare professionals discussed all of the outcome domains that had previously been identified via focus group discussions with patients and carers [11], but with some differences in items within certain domains. However, only visual function (distance vision), treatment side effects and disease control outcomes were assessed in previous clinical research identified as part of the systematic review [10].

Professional interviewees discussed several outcomes related to disease control that were not expressed specifically by patients and carers in the focus group discussions. These included inflammatory grading e.g. anterior segment inflammation, vitreous inflammation, and vitreous haze and retinal inflammation e.g. retinal vasculitis, retinitis. They also pointed out that controlling inflammation (inflammatory grading) helps to prevent the progression of the disease and stop or limit the disease process, therefore, reducing the medication load and hospital visits. It is worth noting however that patients and carers did discuss inflammation at length during focus group discussions and stated that controlling inflammation was a key outcome, even if they didn't use the same terminology as professional interviewees. Within visual function professionals did not discuss depth perception, and within symptoms they did not talk about symptoms such as fatigue and headaches that patients and carers identified. Whilst professionals discussed both of the domains psychological morbidity / emotional well-being and psychosocial adjustment to uveitis, patients and carers emphasised the latter more during

focus group discussions and often focused on these issues more predominantly than professional interviewees.

## 4. Discussion

The aim of this research was to define those outcomes and outcome domains that healthcare professionals considered to be important for non-infectious uveitis of the posterior segment in adult patients, as part of the process of building a multi-stakeholder COS for this condition. The study identified 43 outcomes and grouped these into 11 outcome domains; 1) visual function, 2) symptoms, 3) functional ability, 4) impact on relationships, 5) financial impact, 6) psychological morbidity and emotional well-being, 7) psychosocial adjustment to uveitis, 8) doctor/patient relationships and access to healthcare, 9) treatment burden; 10) treatment side effects and, 11) disease control.

This work is parallel to our recent study eliciting patients' and carers' opinions of the outcomes important in PSIU [11]. Healthcare professionals' views expressed during these telephone interviews were similar to those voiced by patients and carers in their focus groups [11]. The domains developed via the analysis of the focus group data were directly applicable to the data elicited during the interviews with professionals. Healthcare professionals recognised most of the issues discussed by patients and carers in the focus groups. There were some subtle differences in the contents and emphasis of discussion between patients and carers and healthcare professionals. For example, whilst both patients/carers and healthcare professionals identified a wide range of symptoms e.g. pain, red eye, visual disturbance, patients and carers focused a lot on fatigue, whereas healthcare professionals focused more on visual symptoms such as floaters and their impact on patients' functional ability.

Healthcare professionals tended to discuss the disease control and side effects/complications to a greater degree. Despite this, there is a notable concordance when considering the recognition of relevant issues and concepts between patients, carers and healthcare professionals. However, the discordance with outcomes identified in the systematic review highlights the extent to which these issues are not adequately addressed in either clinical practice or clinical trials. A good example is the fact that patients, carers and healthcare professionals all conceived visual function in a multi-faceted e.g. some or all of distance vision, near vision, peripheral vision, colour vision, and contrast sensitivity and holistic manner compared to the relatively limited measures and assessments available and used in clinical practice and research [28–30].

To our knowledge, this is the first piece of qualitative research to explore the views of healthcare professionals on outcomes of importance in the management of patients with PSIU. Although clinical experts' opinions were sought at the Standardisation of Uveitis Nomenclature (SUN) workshop in 2005, the SUN workshop aimed to help clinicians assessing uveitis ocular inflammation and agree standardised grading tools rather than providing a comprehensive list of outcomes in the field. Our qualitative findings present a broader picture of the impact of PSIU and related treatment on patients' lives that was not within the scope of the SUN workshop.

The disease control domain and its associated outcomes were discussed in all interviews. Clinicians' highlight how important to assess the severity of the disease measuring its activity and link those findings to visual function. Healthcare professionals noted that treatment decisions were multi-factorial and included a consideration of the clinical findings including inflammatory markers, structural changes e.g. UMO and visual function and the patient's associated functional ability. Considering all those components would allow a comprehensive assessment of patients' health condition and provide a meaningful dialogue with patients, carers, and other stakeholders involved in the management of PSIU. It is crucial to understand

whether an improvement in disease control outcomes would result in an associated positive impact on patients' lives through improved functionality. For example, a patient is likely to notice a benefit after improved disease control if they have reversible macular oedema but not if that inflammation has become associated with a macular scar.

### Strengths and limitations

This is the first qualitative research study exploring the opinions of healthcare professionals as to how PSIU impacts adult patients, and what outcomes should therefore be measured. Interviews created an in-depth discussion and comprehensive data set that were compatible with the research objectives. No new insights were emerging during the last interview suggesting that data saturation was achieved within this sample [22, 31]. Furthermore, many of the clinicians work with Patient and Public Involvement (PPI) groups therefore may be more familiar with the outcomes that matter to patients through discussion at such groups.

A purposive sample was utilised covering a range of settings in which the healthcare professional worked, their professional group, and their location (although all were from the UK). We recognise a number of limitations to this study: firstly the study did not include healthcare professionals outside the UK. However, most of the healthcare professionals interviewed have some international experience and are part of international expert groups. It is likely that the issues relevant to clinicians practising in uveitis in the UK will have international relevance. Secondly, the number of health policy-makers and commissioners was relatively low. We cannot therefore be sure that further interviews with this group would not identify other domains and outcomes relevant to policy-makers and commissioners, although it was reassuring that data saturation appeared to have been reached. Finally, the interviews were conducted before the COVID-19 pandemic and we are aware that the pandemic has caused disruption to clinical care patterns, and we would anticipate those outcomes that matter to healthcare professionals may remain largely unchanged, however, further research would be required to confirm this.

### 5. Conclusion

This project explored healthcare professionals' perceptions on outcomes important to PSIU patients. These telephone interviews provided rich data on the perceived impact of PSIU covering eleven outcome domains. The data collected from the healthcare professionals were combined with those identified through the systematic review [10], and the qualitative focus groups with patients and carers [11] to provide the long-list of outcomes used to inform a Delphi methodology, the next stage in the development of a COS for non-infectious PSIU in adult patients. Despite the recognition of relevant issues and concepts between patients, carers and healthcare professionals, those findings are not implemented in either ophthalmology clinical practice or clinical trials. This study has value both as contributing to the first COS for PSIU, but also because it provides a unique insight into how health care professionals perceive the impact of this sight-threatening condition and enables comparison to the views of the patients and carers themselves.

### Supporting information

**S1 Table. Questionnaire (interviews).**
(DOCX)

## Acknowledgments

We would like to thank all healthcare professionals who kindly participated in interviews and helped forming the basis of this work.

## Author Contributions

**Conceptualization:** Mohammad O. Tallouzi, Nicholas Bucknall, Melanie J. Calvert, Alastair K. Denniston, Jonathan Mathers.

**Data curation:** Mohammad O. Tallouzi, Philip I. Murray, Melanie J. Calvert, Jonathan Mathers.

**Formal analysis:** Mohammad O. Tallouzi, Jonathan Mathers.

**Investigation:** Mohammad O. Tallouzi.

**Methodology:** Mohammad O. Tallouzi, David J. Moore, Philip I. Murray, Melanie J. Calvert, Alastair K. Denniston, Jonathan Mathers.

**Supervision:** David J. Moore, Philip I. Murray, Melanie J. Calvert, Alastair K. Denniston, Jonathan Mathers.

**Validation:** Philip I. Murray.

**Visualization:** Mohammad O. Tallouzi.

**Writing – original draft:** Mohammad O. Tallouzi.

**Writing – review & editing:** Mohammad O. Tallouzi, Alastair K. Denniston, Jonathan Mathers.

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
