## [Decision Letter · Decision Letter 0]

27 Feb 2023

PONE-D-22-30225Healthcare Professionals’ Views on the Most Important Outcomes for Non-Infectious Uveitis of the Posterior Segment: A qualitative studyPLOS ONE

Dear Dr. Tallouzi,

Thank you for submitting your manuscript to PLOS ONE. After careful consideration, we feel that it has merit but does not fully meet PLOS ONE’s publication criteria as it currently stands. Therefore, we invite you to submit a revised version of the manuscript that addresses the points raised during the review process.

We look forward to receiving your revised manuscript.

Kind regards,

Isabelle Jalbert, PhD

Academic Editor

PLOS ONE

Journal Requirements:

NO authors have competing interests

Additional Editor Comments:

Thank you for the submission. In addition to the requests from your reviewers, please consider the following:

A major limitation of this study is the exclusion from the sample of any of the primary community healthcare practitioners who would be the referrers of patients with non-infectious uveitis of the posterior segment to secondary specialist care such as optometrists and general practitioners. This exclusion needs to be mentioned and justified in the methods and it needs to be acknowledged as a limitation in the discussion of the manuscript. Proper recognition of the potential benefits of interprofessional practice, if only to unsure timely and appropriate referral and reinforce shared-decision making and adherence to complex treatment is warranted is this instance.

Could the authors comment on whether the ordering of the presentation of the domains should be taken to suggest importance or saliency of the domains to healthcare professionals? If yes, could or should the presentation of these domains have been reordered for this paper? For example, it appears that outcome domain 11: disease control was perceived as important by healthcare professionals but it is discussed last in the manuscript. Should you consider re-ordering the presentation of these domains in your result section?

In addition:

1) The use of non standard acronyms should be avoided (e.g., NIU-PS); I note that a different acronym (PSIU) was used in the complementary article published in 2022 in BMJ Open Ophthalmology. To avoid confusion "non-infectious uveitis of the posterior segment" should not be shortened.

2) A sound rationale for the study is provided. The comparation information in Table 3 will provide valuable information to the field and clearly highlights the benefits of conducting qualitative research for identification of core outcome sets.

3) The reporting of the qualitative methods used in this study could be strengthened. For example, did the study follow SRQR or COREQ reporting guidelines (see www.equator-network.org)? A copy of a relevant checklist could be provided as an appendix. A copy of the topic guide used during qualitative interviews should also be available as an appendix. Specify whether the topic guide used in this study was identical to that used in the related study focused on the perspective of patients and their carers?

Reviewers' comments:

Reviewer's Responses to Questions

**Comments to the Author**

1. Is the manuscript technically sound, and do the data support the conclusions?

Reviewer #1: Yes

Reviewer #2: Partly

Reviewer #3: Partly

2. Has the statistical analysis been performed appropriately and rigorously? 

Reviewer #1: N/A

Reviewer #2: I Don't Know

Reviewer #3: No

3. Have the authors made all data underlying the findings in their manuscript fully available?

Reviewer #1: Yes

Reviewer #2: Yes

Reviewer #3: Yes

4. Is the manuscript presented in an intelligible fashion and written in standard English?

Reviewer #1: No

Reviewer #2: Yes

Reviewer #3: Yes

5. Review Comments to the Author

Reviewer #1: General Comments

This manuscript describes the outcome features identified as important by healthcare professionals in the context of non-infectious uveitis of the posterior segment and seeks to see where they align with features identified by patients and carers. There are some grammatical or word choices within the manuscript, particularly in the introduction, which impact readability overall, with some examples excerpted in the specific comments. The information presented on the whole provides insight into perspectives of health care professionals and policy workers of patients with this condition, within the limitations identified by the authors and should be informative for the proposed finish of the Delphi process/study to come.

Specific Comments

Introduction

- Line 97-98: The highlighted sentence: Delete “which”

- Line 106-108: The use of the word “whilst” at the beginning of this sentence suggests that a comparison is going to occur in the sentence which does not occur, rendering the sentence somewhat confusing. Similar grammatical errors in the introduction are seen repeatedly, impacting readability

Methods

- Study Design: please expand on this in this area beyond simply stating that this was “a qualitative research study”

- Line 176: Telephone interviews are introduced as a qualitative research method allowing researchers to have access to a wide geographical area and collect rich and in-depth data with evidence that they can produce equivalent data to face-to-face interviews” Is this sentence a quote or not? There are quotation marks at the end of this highlighted sentence so it is not clear if these are the words of the authors or the reference being cited

- Line 227-228: Please explain what the “(8)” signifies at the end of this sentence: A total of 43 separate outcomes were interpreted and mapped onto the 11 previously reported outcome domains (8):”

Reviewer #2: Thank you for this study on outcomes for practitioner consideration of outcomes for patients with non-infectious uveitis affecting the posterior segment. Please see the following comments and address all issues noted.

1. Abstract line 70 “Healthcare professionals recognize all of the outcome domains as patients/carers….”

Please modify to indicate the patient/carers outcome domains were from a different and published study. Thank you.

2. The Introduction provides the rationale for the study as expected, although the final section does not clearly state that the patient and carer responses were published previously – please include this reference and modify sentence as below. Thank you.

Line 131 “The views of healthcare professionals were combined with those of patients and carers [11], and we now report and compare the perspectives of healthcare professionals the recently reported views of patients and carers with NIU-PS [11].”

3. The details of the practitioner participants are provided in Table 1, but exclusion criteria should be noted. The reasons for not including other eyecare practitioners such as optometrists may also be useful to comment on in the manuscript.

Were the practitioners all urban-based? An explanation of commissioners would also be helpful as this role is not available in all countries.

How many practitioners, commissioners and nurses were contacted for the final number of 12 participants?

The authors also note on line 187:

“Prior to the start of interviews, all participants were asked to complete a short background questionnaire that gathered data on sample characteristics and helped to promote diversity amongst the interview sample.” Please provide details of this additional survey as a supplementary file – what characteristics were surveyed?

4. Were the differences in participant characteristics such as years of experience and clinical trials experience, for example, considered when analysing the findings from the study? How was experience 'measured'?

5. The interviews were conducted in 2018 and wonder if there are concerns as to how the comments from then, may relate to the most recent patterns of clinical care for NIU-PS patients, including what has happened with these patients during COVID? Can the authors please comment and include mention in limitations related to this aspect of their research. Thank you.

6. For the 11 outcome domains included, the authors indicate these were the same domains as those for the patients and carers. In Table 3, these outcome domains and items are then compared for practitioners and patients and carers. Please reference the published study in the column for patients and carers (ref 11). A systematic review is also referred to in Table 3, and this should also be referenced. Thank you.

Minor comments

1. References require attention to detail. Please see examples below (and there may be more). Please check these carefully.

4. Merrill, P.T., et al., Efficacy and Safety of Intravitreal Sirolimus for Non-Infectious Uveitis of the

767 Posterior Segment: Results from the SAKURA Program. Ophthalmology, 2020. No page numbers

9. Coleman, K., et al., Evidence on the chronic care model in the new millennium. 2009. 28(1): p. 75-85. No journal or book is noted; please amend with source reference added.

12. Kan, K., et al., Patients’ and clinicians’ perspectives on relevant treatment outcomes in depression: qualitative study. BJPsych open, 2020. 6(3). No page numbers

28. Birks, M. and J. Mills, Grounded theory: A practical guide. 2015: Sage. Details of edition, publisher required.

Reviewer #3: In this study the Tallouzi and colleagues canvassed a number of clinicians on their views on what they thought were the most important outcomes in the management of non infectious uveitis, then compared their results with systematic reviews and their past work exploring a similar field with patient and carer groups.

Whist it is clear that an extensive amount of work has been done, with a lot of data collected, it is unclear how their data was actually analysed, largely due to how it has been presented to the reader.

In the methods, the authors describe a framework analytical approach that they used in their similar prior work. Although such a framework has been published previously, a brief description in the methodology of this paper should also be provided.

The authors also describe a coding framework of outcome domains, but do not enlighten the reader as to how this coding was done, and thus how the data was actually analysed.

Whilst a comparison of outcome domains with their prior work with patient and carer focus groups is described by the authors, this appears to be a tick box exercise, rather than a rigorous scoring of congruity (or otherwise) within each domain.

One therefore wonders whether the authors have considered analysing their data using a principal component analysis (PCA) to assess the construct validity of their survey questions and determine whether the outcomes were consistent with the desired constructs. The advantage of a PCA is that it is a proven method to simplify the dimensionality of multivariate qualitative data into a limited number of new variables, allowing such multivariate data to be better visualised graphically & therefore easier to analyse overall.

This may therefore reduce the large number of domains presented by the authors in their results, as currently, as presented, what the authors are essentially presenting are a whole lot of quotes grouped into 11 different categories, which is essentially quite hard to summarise into a clear message, particularly when each field is then further subdivided (as per Table 3).

Given the large amount of collected data, the lack of a clear message is a shame, hence the suggestion to consider alternate ways of analysing and thus presenting, these results.

6. PLOS authors have the option to publish the peer review history of their article (what does this mean?). If published, this will include your full peer review and any attached files.

Reviewer #1: No

Reviewer #2: No

Reviewer #3: No

---

## [Author Response · Author response to Decision Letter 0]

16 Apr 2023

We have responsed to the reviewers comments point by point as per attached file

---

## [Decision Letter · Decision Letter 1]

25 Jun 2023

PONE-D-22-30225R1Healthcare Professionals’ Views on the Most Important Outcomes for Non-Infectious Uveitis of the Posterior Segment: A qualitative studyPLOS ONE

Dear Dr. Mohammad Tallouzi,

Thank you for submitting your manuscript to PLOS ONE. After careful consideration, we feel that it has merit but does not fully meet PLOS ONE’s publication criteria as it currently stands. Therefore, we invite you to submit a revised version of the manuscript that addresses the points raised during the review process.

ACADEMIC EDITOR:Kindly provide a response to the reviewer comments and give justifiable reasons for reviewer recommendations that are not achievable. Also include appropriate limitations and provide harmony between this manuscript and the earlier work considering patients and careers perspective to uveitis, this should bear in mind that patients and physicians perspective could be likely divergent and the scale of importance of the domains between the patient and physician may be different.

We look forward to receiving your revised manuscript.

Kind regards,

Ogugua Ndubuisi Okonkwo, M.D.

Academic Editor

PLOS ONE

Additional Editor Comments:

Dear Author,

while, your article has merit, significant issues have been raised by reviewer 3 These issues need to be responded to appropriately in your revision. If you chose not to make the reviewer recommended changes, please give valid reasons for this.

I also, feel that your article can be made more concise and easier to read. Please provide sufficient evidence for the choice of the domains used. The domains are at the heart of the research and the choice should be well explained. Though similar domains have been use in the submission on patients perspective, there is need to validate the use of a similar number of domains for phyisicians, since a patients perspective will be different from a physicians view. Since reference is made to an earlier article publised on the patients perspective, your manuscript should be seen to mirror your previous work, and the terminologies used should be similar to allow easy independent comparison.

Lastly, kindly provide a clear take home of your work, clinical utility of the work done and future directions for work in this area of research.

Reviewers' comments:

Reviewer's Responses to Questions

**Comments to the Author**

1. If the authors have adequately addressed your comments raised in a previous round of review and you feel that this manuscript is now acceptable for publication, you may indicate that here to bypass the “Comments to the Author” section, enter your conflict of interest statement in the “Confidential to Editor” section, and submit your "Accept" recommendation.

Reviewer #1: All comments have been addressed

Reviewer #3: (No Response)

2. Is the manuscript technically sound, and do the data support the conclusions?

Reviewer #1: Yes

Reviewer #3: No

3. Has the statistical analysis been performed appropriately and rigorously? 

Reviewer #1: N/A

Reviewer #3: No

4. Have the authors made all data underlying the findings in their manuscript fully available?

Reviewer #1: Yes

Reviewer #3: Yes

5. Is the manuscript presented in an intelligible fashion and written in standard English?

Reviewer #1: Yes

Reviewer #3: Yes

6. Review Comments to the Author

Reviewer #1: (No Response)

Reviewer #3: As presented, the authors have addressed some of the reviewer comments, but have chosen to ignore others.

Part of the reason according to the author response is that the authors wish to present their findings in the same way as their prior publication. However, the suggestions put forward by the reviewers does not preclude this - they are suggestions that may further enhance the presentation of their results with greater statistical validity, but these have not been directly addressed in this submission

7. PLOS authors have the option to publish the peer review history of their article (what does this mean?). If published, this will include your full peer review and any attached files.

Reviewer #1: No

Reviewer #3: No

---

## [Editor Report · Decision Letter 2]

26 Oct 2023

Healthcare Professionals’ Views on the Most Important Outcomes for Non-Infectious Uveitis of the Posterior Segment: A qualitative study

PONE-D-22-30225R2

Dear Dr. Tallouzi,

We’re pleased to inform you that your manuscript has been judged scientifically suitable for publication and will be formally accepted for publication once it meets all outstanding technical requirements.

Kind regards,

Ogugua Ndubuisi Okonkwo, M.D.

Academic Editor

PLOS ONE

Additional Editor Comments (optional):

Dear author, i have gone through your response to the reviewers comments and found them to be appropriate and satisfactory.
---

## [Editor Report · Acceptance letter]

9 Nov 2023

PONE-D-22-30225R2 

Healthcare Professionals’ Views on the Most Important Outcomes for Non-Infectious Uveitis of the Posterior Segment: A qualitative study 

Dear Dr. Tallouzi:

I'm pleased to inform you that your manuscript has been deemed suitable for publication in PLOS ONE. Congratulations! Your manuscript is now with our production department. 

Kind regards, 

on behalf of

Dr. Ogugua Ndubuisi Okonkwo 

Academic Editor

PLOS ONE